# Fake News Detection Model on Social Media by Leveraging Sentiment Analysis of News Content and Emotion Analysis of Users’ Comments

**DOI:** 10.3390/s23041748

**Published:** 2023-02-04

**Authors:** Suhaib Kh. Hamed, Mohd Juzaiddin Ab Aziz, Mohd Ridzwan Yaakub

**Affiliations:** 1Center for Software Technology and Management (SOFTAM), Faculty of Information Science and Technology, University Kebangsaan Malaysia (UKM), Bangi 43600, Selangor, Malaysia; 2Center for Artificial Intelligence Technology (CAIT), Faculty of Information Science and Technology, University Kebangsaan Malaysia (UKM), Bangi 43600, Selangor, Malaysia

**Keywords:** fake news detection, sentiment analysis, emotion analysis, social media, deep learning

## Abstract

Nowadays, social media has become the main source of news around the world. The spread of fake news on social networks has become a serious global issue, damaging many aspects, such as political, economic, and social aspects, and negatively affecting the lives of citizens. Fake news often carries negative sentiments, and the public’s response to it carries the emotions of surprise, fear, and disgust. In this article, we extracted features based on sentiment analysis of news articles and emotion analysis of users’ comments regarding this news. These features were fed, along with the content feature of the news, to the proposed bidirectional long short-term memory model to detect fake news. We used the standard Fakeddit dataset that contains news titles and comments posted regarding them to train and test the proposed model. The suggested model, using extracted features, provided a high detection accuracy of 96.77% of the Area under the ROC Curve measure, which is higher than what other state-of-the-art studies offer. The results prove that the features extracted based on sentiment analysis of news, which represents the publisher’s stance, and emotion analysis of comments, which represent the crowd’s stance, contribute to raising the efficiency of the detection model.

## 1. Introduction

Fake news affects people’s daily lives [1,2], manipulates their thoughts and feelings [3,4], changes their beliefs [5], and may lead them to make wrong decisions [1]. The propagation of fake news on social media negatively affects society [6,7] in many domains such as political, economic, social, health, technological, and sports [5,8]. A study conducted in 2016 indicated that 23% of people in the US shared fake news, either deliberately or inadvertently [9]. According to a survey, fake news has a 70% greater chance of spreading than real news [5]. Another study has indicated that many people have difficulty distinguishing between fake news and real news, irrespective of their gender, age, or educational attainment [10]. Social media platforms have presented a virtual environment for posting [11], discussion, exchange of views, and global interaction among users [12], without restrictions on location, time, or content volume [13]. A survey conducted in 2017 claimed that 67% of people in the US got their news mainly from social media [14]. In 2021, Facebook announced that about 1.3 billion fake accounts had been closed, and more than 12 million posts containing false information about COVID-19 and vaccines had been deleted [15]. Recent research has shown that the spread of fake news on social networks has become an urgent global problem that must be addressed and curbed quickly as it spreads social panic and even economic turmoil [16,17]. Fake news detection is challenging, and studies on this issue are still in their nascent stage. However, the proliferation of fake news is on the rise, which calls for further development and exploration of new directions in research to improve the techniques used for identifying such news [5,18]. Many studies on the identification of fake news on social networks depend on one or more features such as content, network propagation, or user [19,20,21]. Analyzing users’ comments to ascertain their attitudes toward the news could play a major role in identifying fake news [22,23,24] and giving an idea of the credibility of the published news [14,15]. Albahar [25] posited that user comments have great discriminatory value in the detection of fake news, wherein the expression of sentiment [26] or emotion [27] is crucial. Vosoughi and Roy [28] found that users’ replies to fake news carry the emotions of fear, disgust, and surprise, while users’ replies to real news carry the emotions of anticipation, sadness, joy, and trust; however, they did not examine how well emotions can identify fake news. According to Kumari and Ashok [29], novelty may be a crucial component of fake news and considerably increase its spread and acceptance in society. Novelty grabs people’s attention and stimulates the exchange of ideas and decision making. The majority of existing studies that employ sentiment analysis focus primarily on the content sentiment signals of fake news delivered by publishers rather than the emotion signals of comments provided by crowds [30]. However, fake news frequently generates high excitement or arouses the public’s emotions in order to be widely disseminated among the masses. Therefore, it is important to investigate if emotions in news comments and the relationships among them can aid in detecting fake news in addition to the sentiment present in news content. Furthermore, some users tend to use emojis instead of text comments to express their opinions about certain news on social media [31,32,33]. Therefore, these emojis should be included in the emotion analysis process. In this context, deep learning (DL) techniques have provided a significant advancement in the classification, prediction, and analysis of textual content [34,35]. This is due to their ability to learn effectively [5,36], extract features, and capture complex patterns [37,38]. Research has shown that the bidirectional long short-term memory (Bi-LSTM) models are more efficient than the unidirectional models [39] and that they effectively deal with sequential texts due to their ability to capture long-term dependencies and, over time, store historical and future information [40]. The current research proposes a model using a new approach that employs new trends in the detection of fake news on social networks by utilizing features extracted based on sentiment and emotion analysis, taking care not to neglect the emojis in the comments, as shown in Figure 1, which is a real example of fake news (https://thelogicalindian.com/fact-check/muslim-spit-restaurant-covid-19-coronavirus-20457, accessed on 16 July 2021) based on a report conducted by the BBC (https://www.bbc.com/news/world-asia-india-53165436, accessed on 16 July 2021). This fake news about the Muslim community in India was disseminated in the first week of April 2020; it claimed that Muslims deliberately spread COVID-19. This led to increased hostilities toward Muslims and calls to boycott them economically. The importance of the research lies in the fact that most of the studies that dealt with the matter of emotion analysis-based features of the public’s responses in detecting fake news only provided analysis and statistics of the news in the dataset in terms of containing certain emotions, and these features were not tested and practically utilized in detecting fake news. Therefore, this study presents a model based on a Bi-LSTM classifier to determine whether the news is real or fake. It will use features based on analysis of the sentiments of the news as well as the emotions of the crowd’s comments on this news, utilizing the emojis in the comments. In addition to extracting content features, the extracted features based on the analysis of sentiment and emotions contributed to raising the efficiency of the proposed detection model and provided higher accuracy than state-of-the-art models.

Our contributions could be summarized as follows:Analyzing the sentiments of news titles and also analyzing the emotions of users’ comments on this news and their relationship to fake news in the Fakeddit dataset.Examine the textual content of news headlines in detecting fake news.Investigating the use of sentiment-based features of news headlines, as well as features based on the emotions of users’ comments, to detect fake news.Proposing a model based on Bi-LSTM on both textual features of news headlines and features extracted from both news sentiment analysis and comment emotion analysis to achieve state-of-the-art results for fake news detection.Demonstrating the results on the real dataset (Fakeddit).

The remainder of this study is arranged as follows: Section 2 presents a background on the research, in terms of the fake news effects, the role of social media in spreading news, and the importance of analyzing sentiments and emotions in Natural Language Processing (NLP). Section 3 presents a brief survey of the fake news detection models based on sentiments and emotions analysis, in addition to the review of baseline approaches on the Fakeddit dataset. Section 4 describes the dataset and depicts the research methodology in detail. Section 5 shows the results of the experiments. Section 6 investigates the results of the experiments. Finally, Section 7 concludes this study and suggests future directions for detecting fake news.

## 2. Background of Study

This section is divided into three subsections, as follows: Section 2.1 provides an overview of fake news, its spread, and its consequences, and Section 2.2 discusses the role of social media in the spreading of fake news. Section 2.3 investigates the uses of sentiment and emotion analysis in studies related to the field of natural language processing.

### 2.1. Fake News Overview

“Fake News” has been announced as the Collins Dictionary’s official word of the year for 2017 [14]. Fake news can be defined as published news articles that contain false information to intentionally mislead readers [36] and carry out malicious purposes [20,41]. The spread of fake news on the Internet is faster than the spread of real news [5], because people are curious to know about new information or news [28], and tend to share the latest information [42], especially sharing breaking news without verifying its veracity [41]. Some fake news is unintentionally published or shared and is called misinformation [43]. Repeated viewing of fake news makes it familiar to the recipient, makes it believable, and causes it to be circulated as real news [44]. However, it becomes difficult to distinguish between fake news and real news, as scientific research has revealed that the human ability to distinguish between true and false information is relatively weak, with a rate of about 54% [45]. In addition, research was conducted by researchers at Stanford University, where they observed that students have difficulty determining the reliability of information published on the Internet [4]. A lot of fake news is often related to events or crises that have arisen recently and have not been verified [5,41,42]. However, fake news often disappears from the Internet, including social networks, after some time, but this fake news may have left a deeply negative impact [3,46]. Withholding fake news or rumors without refuting them or revealing the truth may have the opposite effect of making people confused. This may prompt them to speculate or exaggerate the news [1]. Since fake news has become a global challenge and a major threat to democracy, the economy, and peaceful coexistence [47], nongovernmental organizations, civil society organizations, journalists, politicians, and researchers have been working to reduce its risk [5]. As a consequence of this, companies such as Facebook, Twitter, and Google have given significant attention to countering the spread of fake news. These companies have conducted many pieces of research on this aspect [4].

### 2.2. Social Media

The rapid technological development that the world has witnessed in recent years, especially in the mobile industry, has made social media platforms such as Facebook, Twitter, Instagram, and Sina Weibo within reach and an integral part of our daily lives [14,36]. According to the Digital 2021: Global Overview Report released on 27 January 2021, from DataReportal (https://datareportal.com/reports/digital-2021-global-overview-report, accessed on 11 November 2021), as shown in Table 1, the number of people in the world with social media accounts reached about 4.20 billion in 2021, equivalent to more than 53% of the total world population.

Many people nowadays spend their time on social media to connect, get information and news, and share it, rather than watching traditional media [36,39,49]. The main reason for using social media by users around the world is to get news and keep up with current events. This is based on the Digital 2021 Global Digital Overview report, as shown in Figure 2.

Many people use social media networks to post news and information through their accounts or pages because publishing news on these platforms differs from traditional media in that it does not take long, has no cost, and is not subject to audit restrictions [5,14]. The nature of the structure of social media platforms allows the dissemination of news in real-time and quickly, regardless of the credibility of this news [39]. For example, in the United States, in 2012, 49% of users shared news on social networks. Based on a report by the Pew Research Center (https://www.pewresearch.org/, accessed on 23 August 2021), in 2016, more than 62% of users received their news daily from social networks [5], while in 2018, two-thirds of the US adult population received their news from social media platforms [50]. According to the GDATA report, 59% of social network users have been in contact with misinformation [51]. More than 57% of social network users expect this published news to be inaccurate [50]. The Statista website presented a statistic (https://www.statista.com/statistics/649221/fake-news-expose-responsible-usa/, accessed on 4 March 2021) on 27 August 2019 based on a survey conducted in the United States in 2018 about how social networks are responsible for spreading fake news, and the finding of this survey was that 29% of the participants said that social media is mostly responsible for spreading fake news, while 60% of them indicated that these platforms are partly responsible for spreading fake news, as shown in the following Figure 3.

However, despite all these positive features of social networks, there are negative aspects to these platforms, which are exploited by individuals or groups to spread misinformation and fake news for malicious ends [52], which may be for financial gain, to spread hatred based on extremist motives, to manipulate people’s minds for political reasons, or to create biased opinions for electoral reasons [8,53,54]. These negative aspects of social networks, represented by the spread of fake news, portend a serious danger that negatively affects society and the lives of citizens. This calls for the existence of models that detect fake news and limit its propagation [15].

### 2.3. Sentiment and Emotion Analysis in Natural Language Processing (NLP)

Sentiment analysis (SA) is one of the sections that belong to the NLP field [11] and is responsible for designing and applying models, techniques, and approaches to identify whether a text deals with objective or subjective information [55] and, in the last case, to identify whether that information was expressed in a negative, neutral, or positive manner [56] as well as whether it is strong or weak [26]. The SA method is used in many fields, especially in social media [57], such as classifying users’ opinions on social media posts [58,59], or knowing the tendencies of the masses in elections and predicting the final results, in addition to controlling public opinion by understanding the public’s attitudes by analyzing users’ opinions about certain situations [32]. Additionally, it contributes to commercial marketing by exploring the desires of consumers towards the goods offered on social media platforms [60,61,62]. SA is also used to detect fake news, and it is an influential factor in determining misleading information [63], by presenting crucial information about its content. Since a significant portion of the audience of fake news does not read outside the titles, publishers purposely employ combinations of polarity (positive and negative) or emotional valence and excitement (weak and strong) to mislead readers. As a result, headlines should arouse reader curiosity and emotionally attract them in order to increase the spread of fake news [26]. The findings of the research by Paschen and Management [64] show that fake news headlines are significantly more negative than real news titles. This indicates that titles are a powerful sentimental differentiator between fake and real news. Fine-grained sentiment analysis of subjective content can be approached positively thanks to studies in the field [65]. Sentiment analysis usually operates at a coarser level in research efforts. SA focuses more on identifying a text’s subjectivity or semantic position than it does on identifying a particular emotion [66]. Often, it is important to pay close attention to how someone feels after being provoked. For instance, even though sadness and fear are negative emotions, being able to distinguish between them can be crucial. In the event of a disaster, fear may be used to identify the tragedy’s beginning, while sadness may be associated with its final stages [67]. Many previous studies dealt with the sentiment analysis field in social media, but few studies investigated emotions analysis. Emotions analysis is the classification of data concerning the feelings that they carry, such as joy, surprise, anger, fear, sadness, and disgust [33]. Typically, a text contains a number of particular words (that are usually found in emotional lexicons) to convey specific emotions. Therefore, the studies use emotion lexicons that are annotated by experts to extract emotion-based features from the text [30]. There are several emotional lexicons that have been developed that are psychologically well structured, including those put forth by Magda Arnold [68], Paul Ekman, Robert Plutchik [69,70], and Gerrod Parrot [71]. Most of the classifications of emotions found in previous studies derive from the classifications in Plutchik’s model [33], such as those illustrated in Figure 4. This includes the basic classification, which consists of eight categories. The issue of detecting fake news has been studied recently, and the vast majority of these studies only use text-based features. Fake news has the deliberate aim of stirring up readers’ emotions in an effort to be believed and spread on social media. This is one of its defining traits [72]. Emotion analysis plays a key role in determining the user’s behavior toward a specific topic [4]. Vosoughi, Roy [28] presented one study that examined emotions in rumors. They looked at the validity of rumors spread on Twitter, and they discovered that false rumors caused people to respond with fear, disgust, and surprise. In contrast, real rumors caused people to react with joy, sadness, trust, and anticipation.

## 3. Previous Studies

This section discusses relevant studies, as this section is divided into two subsections. Section 3.1 examines the use of features based on sentiment and emotion in detecting fake news. Section 3.2 reviews studies that provide detection models using the benchmark dataset.

### 3.1. Fake News Detection Models Based on Sentiments and Emotions Analysis

Several works related to the detection of fake news have been presented using sentiment analysis-based techniques. Ajao, Bhowmik [73] note that there is a connection between the sentiments of the posted news and the truthfulness of the news, and based on that, they utilized a sentiment-based feature (the ratio of the number of positive to negative words) to support their model in detecting fake news. In order to improve the accuracy of fake news identification, Bhutani and Rastogi [63] have put forth a novel strategy that involves enriching the combined dataset with sentiment as a key feature. The effectiveness of their suggested strategy is also examined using three different datasets, and the results reveal that the proposed solution successfully works. As for the works that used emotional analysis in the detection of fake news, Zhang, Cao [30] confirmed that the difference between fake and real news can be distinguished by dual emotion (emotion of the publisher and societal emotion), and they proposed “Dual Emotion Features” to express dual emotion and the interaction between them. They also showed that their suggested features are simple to integrate as an improvement into already-existing fake news detection models. After analyzing the data, the researchers found that there is a statistically significant difference between real and fake news based on the emotions of the publishers and social emotions. In fake news, there is a lot of emotional inconsistency, such that the publisher’s emotions convey an emotion of happiness, while the social feelings are anger. Many experiments have been conducted by Zhang Cao [30] on three real-world datasets—two in Chinese and one in English—which illustrated that their suggested features set: (1) performs better than the recent emotion-based features related to the task; (2) could be easily integrated with current fake news detection models and significantly enhance the performance of detecting fake news models. Giachanou, Rosso [72] suggested the EmoCred strategy, a long short-term memory (LSTM)-based model that takes into account emotional cues to distinguish between real and fake claims. EmoCred’s key step is to extract emotional signals from the claims. The researchers investigated three distinct methods for determining the claims’ emotional cues: (i) a method that incorporates modern emotion lexicons and is lexicon-based (emoLexi); (ii) a method that determines the intensity of emotions of the claims (emoInt); and (iii) a neural network (NN) method that specifies the level of intensity of the claims and that represents the number of emotional reactions that the claim could elicit in the readers (emoReact). Their study, which used real-world datasets, demonstrated the value of emotional cues in the assessment of credibility. Kumar, Asthana [4] classified the news into real and fake based on emotion analysis of news stories by using an ensemble model consisting of convolutional neural networks (CNN) and BI-LSTM networks with an attention mechanism. The results of their ensemble model were superior to those of the other used models, and they observed that fake news carries exciting emotions, while real news carries monotonous emotions.

### 3.2. Baseline Models

Most of the previous studies provided fake news detection models that relied mainly on content features; some of these models employed network or user-based features. Most of these studies proposed models based on deep learning since the latter is efficient at extracting features [74,75]. Few studies have employed extracted features based on sentiment analysis for news and sentiment analysis for user comments. Lack of a dataset containing the comments of social media users on the news is one of the reasons for not using the method of analyzing the emotions of public responses. In this section, we will review studies that used the Fakeddit dataset to identify fake news. This dataset contains users’ comments, which can be used in our research. The performance result of our proposed model will be compared with these studies in the experimental results and discussion section. Nakamura, Levy [76], who created the multimodal Fakeddit dataset, applied the BERT model to detect text-based fake news, and they utilized the ResNet technique to detect image-based fake news. The researchers found that the accuracy of the model results based on BERT and ResNet in using features of both text and images was better than the model that uses only text in detecting fake news. Kaliyar, Kumar [77] suggested a detection model for fake news that relies on DeepNet by using the real data of BuzzFeed and Fakeddit datasets. By utilizing tensor factorization, which integrates news content and social context-based data, DeepNet outperformed current fake news detection models. The outcome demonstrated that employing a combination of social context-based features and news content features led to more accurate Deep-Net results. Kirchknopf, Slijepčević [78] also used the Fakeddit dataset and presented a multimodal network architecture that permits various levels and kinds of information fusion, incorporating not only the text of titles but also metadata and other content pertinent to the news titles. Their methodology for identifying misleading information depends on four input modalities: (i) primary textual content of the news or post; (ii) secondary information, or reaction to primary content (such as comments); (iii) the visual content of the post; (iv) any other available metadata. To take into account the unique fundamental structure of the modalities, they combined information at various levels. The performance of their model was enhanced by the additional modalities, proving that they provide valuable information. A unique stance extraction and reasoning network, known as SERN, as well as a sentence-guided visual attention mechanism, were developed in a study by Xie, Liu [79] for multimodal fake news detection. They explained that the responses submitted by various readers comprise both facts and some conclusions based on accepted knowledge. Therefore, they used stance through a graph-based reasoning network and implicitly modeled it, which saved a lot of time and effort. The researchers used the Fakeddit and PHEME datasets for their study, and they indicated that their proposed model provided encouraging results compared to other state-of-the-art models. Raza and Ding [80] developed an approach that utilizes features of news content and social context-based features to detect fake news. Their approach is based on the transformer architecture, which is built from two blocks: an encoder block for extracting meaningful representations from the fake news data, and a decoder block for predicting future behavior based on past data. They performed extensive trials on the data from the two real-world datasets, Fakeddit, and NELA-GT-2019, to assess the efficacy of their proposed approach. They employed the under-sampling strategy, which involves eliminating records from the majority class in order to bring the majority class closer to the minority class, to address the issue of data imbalance in both datasets. Raza and Ding [80] have utilized the social context features of the Fakeddit dataset. The findings of their model using data from both datasets, Fakeddit and NELA-GT-19, suggest that in order for their model to operate at its optimal level, both the content of news and social contexts must be taken into account. Since our proposed model is based on text posts (news titles and users’ comments), our model will identify fake news based on the textual content of news titles, in addition to using features based on sentiment analysis of titles and emotion analysis of users’ comments. Therefore, posts without comments will be excluded in addition to neglecting the image feature, and other features, because we want to test the impact of sentiment analysis-based features and emotion analysis-based features associated with textual content.

## 4. Materials and Methods

This section highlights the dataset used in this study; in addition, it presents in detail the proposed methodology for providing the fake news detection model. This section consists of two subsections. Section 3.1 addresses the dataset used, and Section 3.2 contains the proposed news detection model.

### 4.1. Dataset

This section, which deals with the dataset used in this research, is divided into two subsections. Here, a description of this dataset is provided in Section 4.1.1, and it is also analyzed and visually represented throughout Section 4.1.2.

#### 4.1.1. Dataset Description

The Fakeddit (https://github.com/entitize/Fakeddit, accessed on 22 February 2022) dataset is a large-scale and multi-modal dataset (text and image) collected from the social media Reddit platform for the period from 19 March 2008 to 24 October 2019, by researchers Nakamura, Levy [76]. This dataset consists of more than one million posts from various domains. Several features are associated with these posts such as images, comments, users, domains, and other metadata as shown in Table 2. This dataset contains a lot of noise and null values. A single post may contain multiple comments and may not have a comment. For each post, the researchers provided three labels, classified in two ways, three ways, and six ways.

Fakeddit is the benchmarking dataset of our research, which is a cutting-edge multimodal dataset presented by Nakamura, Levy [76]. They had created hybrid text and image-based models and conducted in-depth experiments for numerous classification variations, highlighting the significance of the innovative multimodality and fine-grained classification features exclusive to Fakeddit.

#### 4.1.2. Dataset Visualization

The Fakeddit dataset contains a lot of noise or duplicate news titles. In addition to that, many records in the title column do not contain content. Some of these titles do not contain comments. Since our proposed model is a binary classification, we will exclude any news that does not have binary labels. Thus, the size of the dataset used in this research in its current format consists of 22,788 records. This is after removing noise, duplication, and records that do not contain a title or titles that do not contain comments. This news is divided into two classes: 8553 instances of fake news and 14,235 instances of real news, meaning this dataset is imbalanced, as shown in Figure 5. Since we deal with sentiment analysis of news in addition to emotion analysis of comments on this news, and for the sake of credibility and to not be biased in favor of the model, we did not balance these two classes by using some techniques such as under-sampling or over-sampling, so that the ratio of the results for analyzing the sentiment and emotion of the text used, which are related to fake and real news, did not change.

Based on TextBlob (https://pypi.org/project/textblob/0.9.0/, accessed on 2 March 2022), which is a lexicon-based sentiment analyzer, the sentiment of the titles in this Fakeddit dataset was analyzed. Sentiment polarity is the output that TextBlob produces after receiving a title. The output represents the polarity that falls between (−1 and 1), where −1 indicates a negative sentiment and +1 indicates a positive sentiment, and between them, a value of zero denotes a neutral sentiment. The results of analyzing the news titles from the dataset used in this study, as shown in Figure 6, using the mentioned technique are consistent with what has been reported in previous studies, which showed that fake news titles tend to be negative more than positive, but real news titles tend to be positive, although most of the titles in both classes are moderate.

Regarding the analysis of emotions of comments, we used EmoLex (https://pypi.org/project/NRCLex/, accessed on 5 March 2022), which is a lexicon-based emotion analyzer from the National Research Council Canada (NRC) [81], to analyze the emotions of comments into eight categories of emotions, which are anger, fear, anticipation, trust, surprise, sadness, joy, and disgust. Previous studies [28,29] indicated that the majority of comments on fake news carry emotions of fear, disgust, and surprise, while most comments on real news carry emotions of anticipation, sadness, joy, and trust. We have gathered the comments that carry emotions of fear, disgust, and surprise, more than the emotions of anticipation, sadness, joy, and trust in the “Novelty” group, either comment that carries the emotions of anticipation, sadness, joy, and trust, more than fear, disgust, and surprise, we gathered them in the “Expectation” group, as for the comments in which the emotions of the “Novelty” group were equal to the emotions of the “Expectation” group, we included them in the “Neutral” group as shown in Table 3. In addition, to make model training less sensitive to the scale of features, data normalization for the emotion groups is performed by setting the values between 0 and 1. This enables our model to converge to better weights, which results in a more accurate model. The result of analyzing the emotions of the comments in the Fakeddit dataset, as illustrated in Figure 7, was consistent with the previous studies mentioned above, in that the majority of the comments on fake news carried the emotions of the “Novelty” group, while the comments on real news carried the emotions of the “Expectation” group more than they carried the emotions of the “Novelty” group.

Figure 8 represents the final format of the dataset used which consists of the “id”, “title”, “comment” and “label”, as well as the “sentiment polarity” and “emotion group” resulting from the analysis of the sentiments of news and the analysis of the emotions of the comments, respectively, which was combined with the main dataset that will be fed to the fake news detection model.

### 4.2. The Proposed Detection Model

The proposed model for detecting fake news is divided into three sub-units, namely: emotion analysis unit, sentiment analysis unit, and text classification unit, which are merged by the concatenation layer, and sigmoid layer for final prediction, as illustrated in Figure 9. This first unit, which is a sentiment analysis unit, analyzes the sentiment of the news titles, while the second unit, a text classification unit, classifies the news based on the textual features of the titles, and finally, the third unit, an emotion analysis unit, analyzes the emotional state of the comments. To clarify more, the first and second units process the news titles in the used dataset, and the third unit processes the comments on this news in the same dataset. For the news sentiment analysis and comment sentiment analysis unit, we will add two columns to our dataset. We will save the sentiment and emotions values that were predicted by analyzing text by the lexicons in these two units. Added these features play an essential role for us because it enriches the dataset by adding other features to this dataset that were derived from the original dataset. We hypothesize that the sentiments and emotions that are put forth in writing news titles or comments could be a deciding factor in the task of identifying the news as fake or real. For instance, when someone makes untrue accusations against another person or makes a false statement about another person, the sentiments that these claims carry are generally negative, which makes it easier to identify them as fake. Additionally, surprising reactions to fake news that are implausible help identify them as fake. As a result, it will further improve the accuracy of the detection model by understanding the sentiments and emotions associated with this news. Bi-LSTM models capture the sentence information from two different perspectives, which leads to greater performance [9] in many tasks, such as text classification [39]. Therefore, we implemented our proposed model based on Bi-LSTM and conducted our experiments in Python language on the Google Colab platform (https://colab.research.google.com/, accessed on 11 March 2022). This proposed model consists of three subsections, which are the following: 4.2.1 emotion analysis unit, 4.2.2 sentiment analysis unit, and 4.2.3 text classification unit.

#### 4.2.1. Emotion Analysis Unit

By analyzing the emotions of the crowd’s reactions to the news, the unit takes into account the crowd’s position on the news. Previous studies indicated that analyzing the emotions of crowds’ comments contributes to the detection of fake news [30]. Based on the dataset analysis, as well as the related works [28,29], the majority of comments about fake news have emotions (fear, surprise, disgust), while users’ comments about real news have emotions (anticipation, trust, sadness, joy). This unit will use the dictionary of emotions to classify comments into eight categories of emotions: anger, fear, anticipation, trust, surprise, sadness, joy, and disgust. Many users use emojis to express their feelings on social media [31]. Emojis are one of the communication methods between users, are a form of visual language that expresses feelings across cultures, and have been growing in popularity lately [82]. Currently, many comments on social media contain emojis. This is because these emojis can fill the gap when expressing textual feelings [82]. Due to their frequent use, some researchers have included them in the sentiment analysis model [32]. Accordingly, due to the importance of emojis, they will not be deleted. Instead, these emojis will be converted into text and replaced with words that correspond to their meaning. This will be conducted using a dictionary that contains 2387 emojis (https://drive.google.com/file/d/1G1vIkkbqPBYPKHcQ8qy0G2zkoab2Qv4v/view, accessed on 19 April 2022) as shown in Figure 10.

Since the comments in social media represent colloquial language because they are related to the public and contain a lot of noise, the comment text will be processed using pre-processing techniques, which are, respectively:Remove new lines and tabs.Striping HTML TagsRemove the linksRemove accented characters.Expanding contractionsRemoving special characters except (!, ?)Reduce repeated charactersCorrecting misspelled wordsRemove stopwordsRemove whitespaces

After the noise has been removed from the comments, this unit uses NRCLex to classify comments based on the emotions they carry into eight categories of emotions. NRCLex is a project (https://pypi.org/project/NRCLex, accessed on 15 April 2022) that has received approval from MIT (https://www.mit.edu/, accessed on 15 April 2022) and predicts two sentiments (negative and positive) and eight basic categories of emotions (anger, fear, anticipation, trust, surprise, sadness, joy, and disgust) in a given text. This package, which is built on the WordNet (https://wordnet.princeton.edu/, accessed on 17 April 2022) synonyms set from the NLTK library and the affect lexicon of the National Research Council of Canada (NRC) (http://saifmohammad.com/WebPages/NRC-Emotion-Lexicon.htm, accessed on 15 April 2022) has over 27,000 terms. Before listing the steps for extracting features based on emotions using NRCLex, three groups will be created according to their relationship and contribution to the detection of fake news, as mentioned in the Dataset Visualization section, and the groups are:The novelty group contains the emotions of (fear, disgust, and surprise).The expectation group contains the emotions of (anticipation, sadness, joy, and trust).The neutral group, in which the emotions of the novelty group are equal to the emotions of the expectation group.

Finally, we summarize the steps for extracting emotion-based features from comments in this unit with the following points:Analyzing each comment according to the emotions it carries based on NRCLex.Categorizing each comment to the group it belongs to (novelty, expectation, and neutral) based on emotional polarity.Grouping the comments for each piece of news based on ID.Assign the largest group (most frequent group) of all comments for each piece of news. In the case of equality between the two groups (novelty, expectation), we assign the neutral group to the news.

Since the model only deals with numeric data, we need to do data normalization for the emotion groups and convert the three resulting groups into numbers between 0 and 1, (0 = expectation, 0.5 = neutral, 1 = novelty). The generated column represents the emotion-based feature according to the groups proposed in this paper that will be combined with other features in the concatenate layer.

#### 4.2.2. Sentiment Analysis Unit

The SA unit analyzes the sentiment of news titles. This unit uses a lexicon-based sentiment analyzer, which is TextBlob, to analyze sentiments after the text has been processed using pre-processing techniques, which are: (1) removing empty lines and tabs, (2) stripping HTML tags, (3) removing links, (4) removing accented characters, (5) expanding contractions, (6) removing special characters except (!, ?), (7) removing stopwords, and (8) removing whitespaces, respectively. It should be noted that the pre-processing steps were applied to the same text entered into the sentiment analysis unit and the news classification unit, as shown in Figure 9. TextBlob is a Python-based NLP library that uses the Natural Language Toolkit (NLTK). The result of TextBlob is polarity, and the polarity score ranges from (−1 to +1), where ( + 1) refers to the most positive terms, such as “great”, and “best”, and (−1) refers to the most negative words, such as “disgusting”, “terrible”, and “pathetic”. TextBlob receives one sentence as text input, and the sentence will often be represented by a collection of words. Following the individual scoring of each word, the ultimate sentiment is determined using a pooling procedure by averaging all the sentiments. The results of the titles sentiment analysis will be used as a feature and will be combined with the rest of the features resulting from other units in the concatenate layer to contribute to detecting fake news within the proposed model.

#### 4.2.3. Text Classification Unit

This unit extracts text-based features of labeled examples of fake and real news using a Bi-LSTM classifier. Firstly, this unit receives news titles after being processed using the pre-processing techniques mentioned in the previous unit (sentiment analysis unit), and after processing them, they are divided into two sets, one for training the classifier and making up 80% of the total dataset, and 20% for the test set. The text is then tokenized and padded. Secondly, we will employ a pre-trained glove as a word embedding model for better representation of the titles and learn the contexts of the words in the training set. The word embedding method is a type of document representation that represents words and distributes them into vectors semantically and syntactically [59]. Embedding converts each word in the text string into vectors of *n* dimensions, where *n* is the embedding dimension, which represents 300d in this proposed model. The distance between the two embedding vectors indicates the proximity of the words in terms of their relationship to each other [39]. For example, the words “anxiety” and “depression” are semantically related because they belong to the same category related to mental health [40], and also the words “bad” and “good” are embedded closely [83]. We used the cased pre-trained GloVe model (840B tokens, 2.2M vocab, 300d vectors) (https://nlp.stanford.edu/projects/glove/, accessed on 10 March 2022) in our model to take both cases for both uppercase and lowercase writing. It is beneficial in detecting fake news because capitalized words are frequently used, and the size of the vocabulary is another crucial factor [84]. Finally, after each title is represented as a vector of words using the pre-trained Glove, the sequences of vectors are then used as inputs to the Bi-LSTM classifier one by one. The results in several tasks such as text classification showed that the bidirectional-LSTM model was more efficient than the unidirectional models [39], where the Bi-LSTM model captures sentence information from two different sides, which leads to higher accuracy in performance. The Bi-LSTM classifier consists of two layers of long short-term memory (LSTM), as illustrated in Figure 11, the forward LSTM layer, and the backward LSTM layer, which combine long periods of contextual information from both directions, front and back, for a certain period [4]. The structure of the BLSTM enables the capture of the greatest number of salient features from both directions [85]. The forward layer learns the sequence of the inputs provided. This information is processed by the forward LSTM from left to right. Its hidden state can be illustrated by following the formula:(1)$$\vec{h}t=\mathrm{LSTM}(x_{t\;},\vec{h}t\;-\;1)$$

The backward layer learns the reverse of the sequence of those inputs; that is, the information will be processed by the backward LSTM from right to left, and its hidden state can be represented as the following formula:(2)$$\overset{\leftarrow }{h}t=\mathrm{LSTM}(x_{t\;},\overset{\leftarrow }{h}t\;+\;1)$$

These two layers of LSTMs are connected to the single output layer as shown in Formula (3). They pass through the input sequence from two different directions at the same time.
(3)$$h_{t}\;=\;[\vec{h}t,\;\overset{\leftarrow }{h}t]$$

Proper hyperparameter selection is a crucial step in every deep-learning solution. As indicated in Table 4, our Bi-LSTM model’s hyperparameters were selected based on the highest results after we tested and compared the outputs of many classifiers using different learning paradigms (different optimal hyperparameters and architectures) [86]. Since the imbalanced dataset leads to overfitting in the performance of the model [87,88,89], consequently, the proposed detection models faced overfitting in their performance because the dataset used in this research is imbalanced (we did not take action to balance the dataset for the reasons referred to in the “Dataset Visualization” section). To prevent the performance of the proposed model from overfitting as well as to increase the generalization of the model to the new data [90,91], and to generate high detection accuracy [92], we used several techniques and methods, including reducing the complexity of model design [93,94], using weight regularization with L1 and L2 regularizers, where L1 is the sum of the absolute weights, and L2 is the sum of the squared weights [90,93], and adding a dropout layer [91,92]. Surely, every piece of news will be dealt with according to its characteristics, extracted based on the analysis of sentiment and emotions, and according to its unique ID. By using the Bi-LSTM model and the concatenation layer, other extracted features will be combined with the text features derived from the Bi-LSTM model, and by using the sigmoid layer, fake news will be identified. This section contains two subsections: (i) a concatenation layer, and (ii) a dense layer.

i.Concatenation Layer

In this layer, the emotions feature and the sentiment feature generated from the previous two units will be combined with the output of the Bi-LSTM layer in the text classification unit to be sent to the next layer (sigmoid layer) within the proposed, model as illustrated in Figure 12, where the NLP input represents the textual features of the news title and the meta input represents the two features of emotions and sentiment.

ii.Dense Layer

Finally, the output of the concatenation layer is passed to the dense layer using a sigmoid activation function. This produces the final output of the proposed model for labeling news as real or fake. This sigmoid function generates the resulting values between 0 and 1, and its formula is Equation (4).
(4)$$\emptyset \left(x\right)=\frac{1}{1+e^{-x}}$$

### 4.3. Performance Evaluation

Since we deal with an imbalanced dataset, in order to evaluate the performance of the proposed models, the area under the curve (AUC) performance measure was used. The area under the ROC curve, referred to as AUC, is a measure that is used to compare learning algorithms and build optimal learning models. AUC near 1 indicates a superior system that can accurately discriminate between real and fake news, whereas an AUC close to 0 indicates a weak system (that is, the system will consider all fake news as real and all real news as fake) [26]. For example, if the AUC is 0.9, this means that the model is 90% able to distinguish between negative and positive classes. AUC is typically used in imbalanced classification tasks [13], such as fake news detection. In this task, the distribution of ground-truth fake news titles and real news titles is significantly imbalanced. Therefore, AUC is better than accuracy in terms of statistical consistency and discrimination. Accordingly, we used the AUC measure as illustrated in Equation (5), as well as the F1-score metric as shown in Equation (6), and accuracy to evaluate the proposed models.
(5)$$\mathrm{AUC}\;=\frac{1-\mathrm{FPR}+\mathrm{TPR}}{2}$$

The true positive rate, or TPR, is an acronym that stands for the percentage of positive examples that are successfully classified. In contrast, FPR, which indicates a false positive rate, is the ratio of cases that were incorrectly classified as negative to all other instances [95]. For an F1-score, the following Formula (6) shows how the F1-score is calculated.
(6)$$F1-\mathrm{Score}=\frac{2\times \;\mathrm{Precision}\;\times \;\mathrm{Recall}}{\mathrm{Precision}+\;\mathrm{Recall}}$$

The harmonic mean of precision and recall yields the F1-score, where the precision of a class of predictions is the proportion of correctly classified as positive cases to all other positive cases. The percentage of correctly classified positive examples is determined using recall. In terms of accuracy, it is the percentage of correctly classified instances over all instances.

## 5. Results

To obtain a model that provides high accuracy in detecting fake news, various aspects that affect the model’s performance were investigated. These aspects include the features chosen, model selection, choice of hyperparameters, and the structure of the model. We carried out the experiments on the Google Colab platform, which is based on the Python programming language. We employed several libraries in the implementation such as Pandas, NumPy, Sklearn, Gensim, and Keras; the codes for the models used in this research are available on our account on the Google Colab platform (https://colab.research.google.com/drive/1edXYIghmzu3Bs4UhWJ8Ho9sk13oaebfz?usp=sharing, accessed on 11 December 2022). We tested several efficient DL models such as LSTM, gated recurrent unit (GRU), Bi-LSTM, and CNN; each model was tested in two stages. In the first stage, models were used only to test the text features of the titles, while in the second stage; the models were applied to test the sentiments of the titles and the emotions of the comments resulting from the two units of sentiment analysis and emotions analysis, as well as to test the effectiveness of text-based features of news titles. Note that all the proposed models used the most effective structure and hyperparameters that resulted from multiple experiments to reach high detection accuracy and the best performance of the model. Table 5 includes the results of the proposed detection models according to the features examined, as measured by the AUC and F1-score metrics. Note that these results were obtained through experiments conducted on validation sets.

Based on the results mentioned in Table 5, the top-performing detection model using the validation set was Bi-LSTM, with an AUC value of 96.77% and 97.81 based on the F1-score. Figure 13, Figure 14 and Figure 15 represent the structure of the proposed Bi-LSTM model, the training and validation AUC of the model, and the training and validation loss of the model, respectively.

The following Table 6 shows the features used by the proposed models in detecting fake news in related works that used the same standard dataset as compared to our model.

Table 7 illustrates the results of our proposed model based on Bi-LSTM in comparison with models presented in relevant studies, using several measures such as AUC, accuracy, and F1-score.

## 6. Discussion

One of the challenges that we faced, is that the models fell into overfitting during the training in the second stage, i.e., when adding numerical features (the values of sentiment features of news and emotion features of comments), due to the dataset used being imbalanced, as mentioned earlier in the text classification unit section. This issue was overcome by adding a dropout layer and using regularizers L1 and L2 for both LSTM and GRU models in the second stage, while only regularizers L1 and L2 were used with the Bi-LSTM model in the second stage. Further, for the CNN model, the batch normalization technique was used in addition to the use of dropout layers in the second stage. The batch normalization layer enables network layers to learn more independently, as the output of previous layers is normalized using this layer. In addition to improving learning efficiency, single-batch normalization can be employed as regularization to prevent overfitting the model [92]. It can be applied in some places between the model’s layers; it is frequently used, especially after the convolution and pooling layers. In addition, an uncomplicated and simple-layered network was used to avoid overfitting. Regarding the results, we observe, as shown in Table 5, the accuracy of fake news detection for all models used has increased relative to the AUC metric when using features based on sentiment and emotion analysis in addition to the text features. This indicates that feature extraction plays an important role in raising the performance efficiency of a model, not only in machine learning models but also in DL models, if these features are more effective when dealt with separately. Therefore, these features must be extracted and handled separately from the text content. The performances of the Bi-LSTM and CNN models were close in terms of results, although the time taken to train the Bi-LSTM model was much longer than that of the CNN model. Despite the fact that some of the related works that were reviewed previously in the Related Works section utilized image-based features and meta features in addition to text features, as shown in Table 6, our proposed model provided a higher accuracy result than the related works, as illustrated in Table 7. This is due to the fact that in this research, new important features were explored and utilized in detecting fake news, and our proposed model provided a high efficiency in detecting fake news compared to relevant studies that used the same benchmarking dataset. Sometimes features used in other studies may not provide a significant improvement in the performance of the detection models, or they may contain noise that reduces their performance. Sometimes, the auxiliary features used in previous studies may not provide a significant improvement in the performance of the model, as a study Kim, Kim [21] indicated that the accuracy of identifying rumors using content features only was higher than using all features combined at the same time. In the future, we plan to analyze the emotions embedded in news and find out if fake news contains certain emotions that represent the publisher’s stance, in addition to analyzing the emotions of user comments by exploring other types of emotions that represent the public’s responses to such news.

## 7. Conclusions

The spread of fake news on social networks is one of the major problems of our time. It has negative effects on society and the lives of citizens. In this paper, many studies aimed at curbing fake news dissemination have been presented, and the majority relied on the features of textual content. Moreover, many studies have used features based on the social network or the user, while others have provided a multimodal approach that combines textual and image features for detecting fake news. In this research, the public’s attitudes toward news were taken into consideration by analyzing the emotions displayed in their comments. Since the majority of comments toward fake news carry emotions such as fear, disgust, and surprise, while comments toward real news carry emotions such as anticipation, sadness, joy, and trust, these emotions were employed as features to identify fake news. Furthermore, many comments on social media contain emojis that depict the user’s emotions; thus, these emojis were not deleted but were replaced with the words they represent. In addition, fake news often carries negative sentiment and represents the publisher’s stance; consequently, news sentiment was analyzed and used as a feature in detecting fake news. Experiments have shown that adding sentiment-based and emotion-based features increases the accuracy of fake news detection for all proposed DL models compared to the use of only text-based features. Our proposed Bi-LSTM model provided the most accurate results in comparison with related works that used a benchmarking dataset. Based on the findings, features based on sentiment analysis of the news and emotion analysis of users’ comments on this news can be employed by social media platforms in combating the spread of fake news. However, this study faced difficulties in dealing with an imbalanced dataset. In the future, the issue of an imbalanced dataset could be addressed through GAN techniques. In addition, we are looking forward to achieving better accuracy by utilizing other state-of-the-art models, such as transformer-based models.

## Figures and Tables

**Figure 1 sensors-23-01748-f001:**
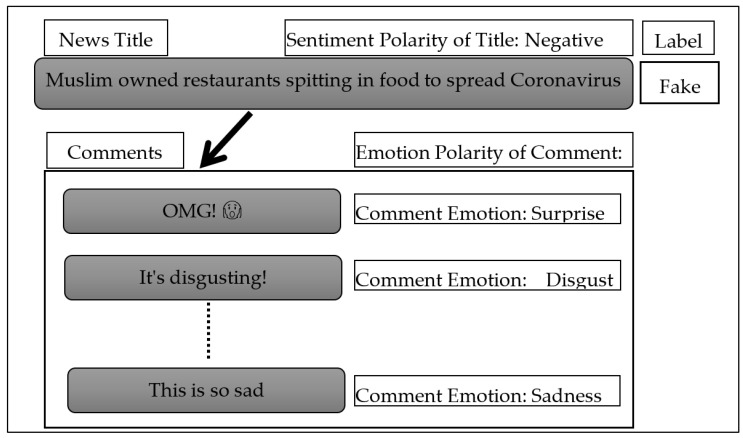
The sentiment analysis of fake news and emotion analysis of the public’s comments about fake news, (These comments are the reaction of the users toward the title of the posted news).

**Figure 2 sensors-23-01748-f002:**
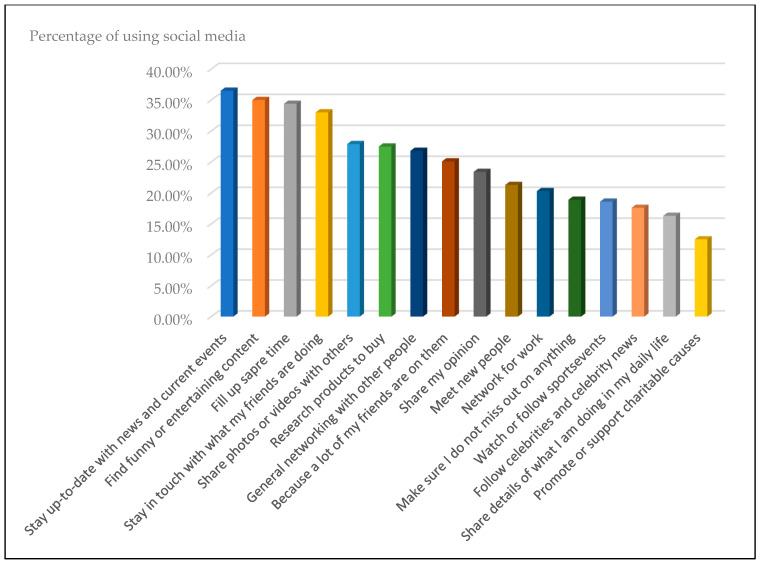
The reasons for using social media according to the Digital 2021 Global Digital Overview report [48].

**Figure 3 sensors-23-01748-f003:**
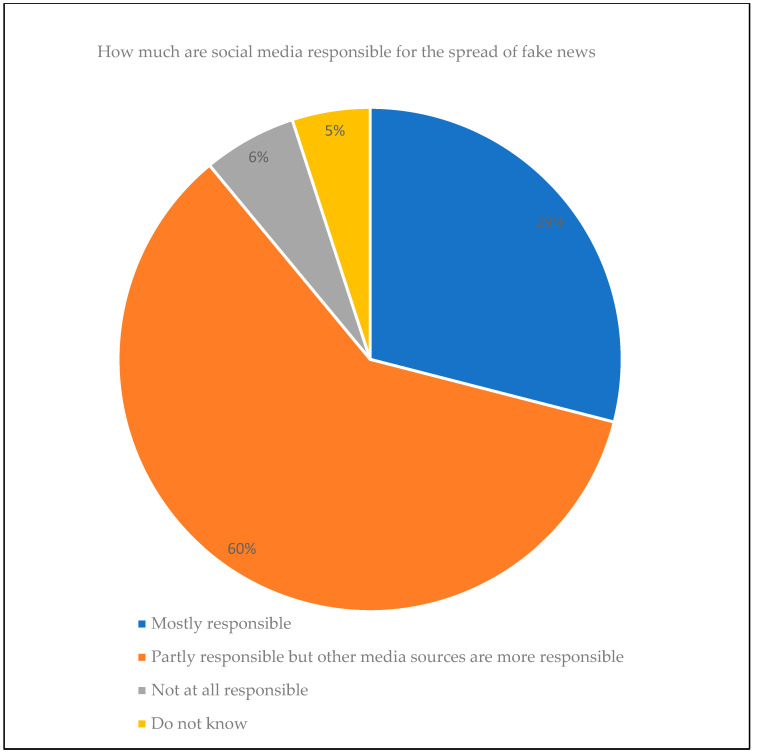
Social media’s role in spreading fake news (Available from: https://www.statista.com/statistics/649221/fake-news-expose-responsible-usa, accessed on 4 March 2021).

**Figure 4 sensors-23-01748-f004:**
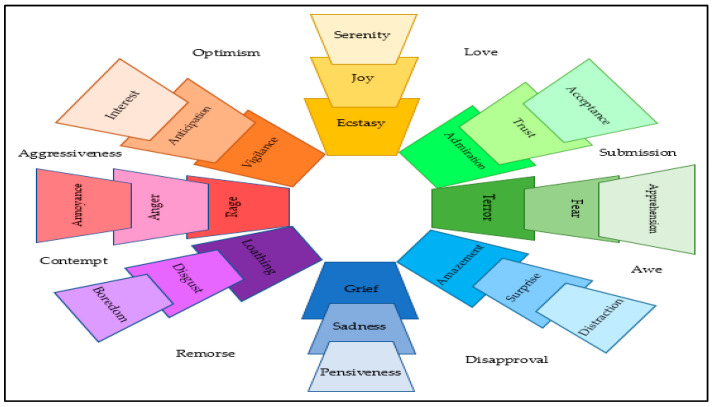
The Plutchik wheel of emotion [33], (This wheel divided into twenty-four primary, secondary, and tertiary dyads that represents the eight emotions of joy, trust, fear, surprise, sadness, disgust, anger, and anticipation).

**Figure 5 sensors-23-01748-f005:**
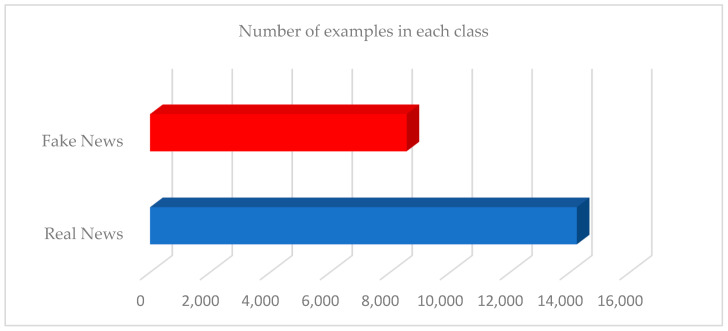
The size of the Fakeddit dataset.

**Figure 6 sensors-23-01748-f006:**
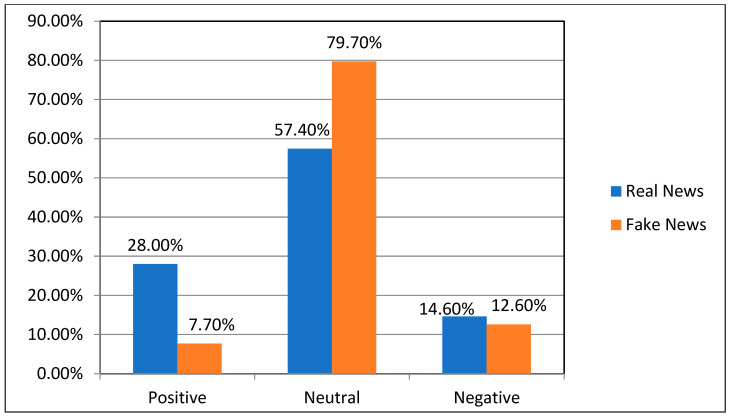
The polarity of sentiment analysis of news titles in the Fakeddit dataset.

**Figure 7 sensors-23-01748-f007:**
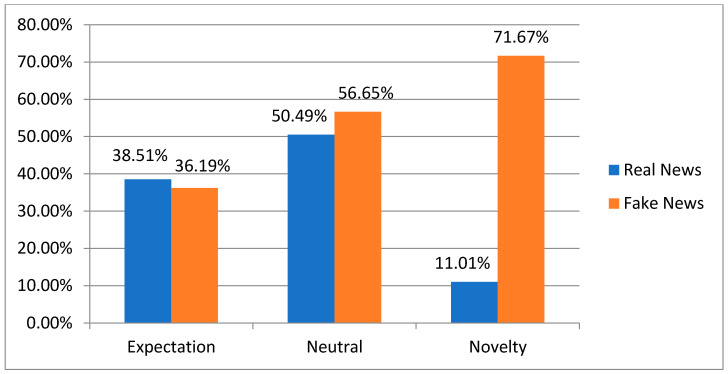
The groups of emotion analysis of comments in the Fakeddit dataset.

**Figure 8 sensors-23-01748-f008:**
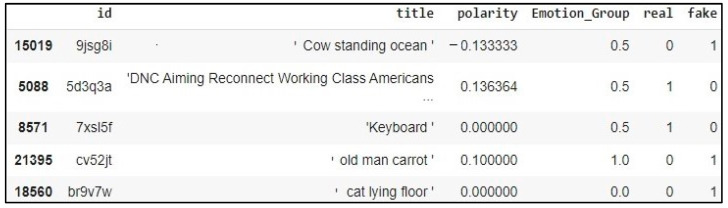
The screenshot of the final dataset used in the proposed model.

**Figure 9 sensors-23-01748-f009:**
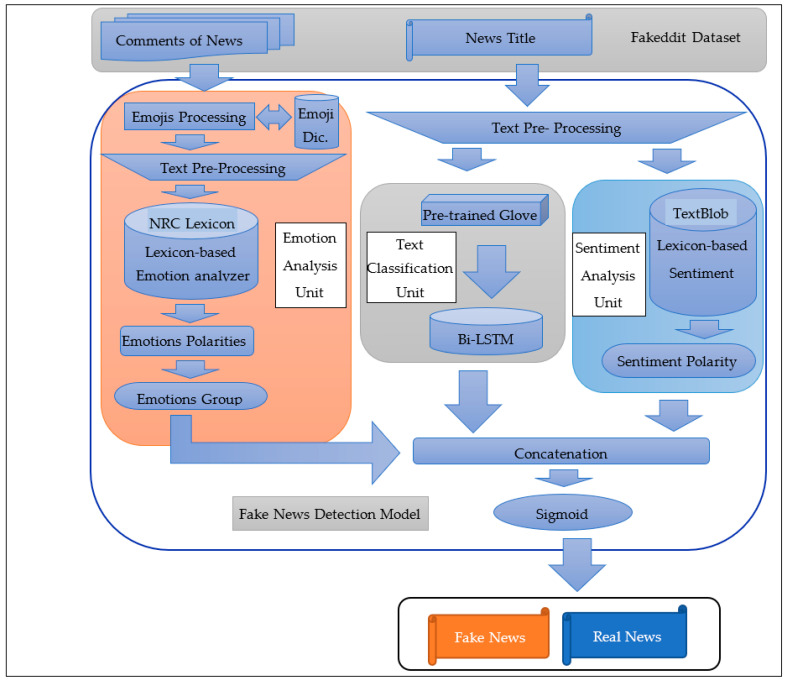
The general design of the proposed model.

**Figure 10 sensors-23-01748-f010:**
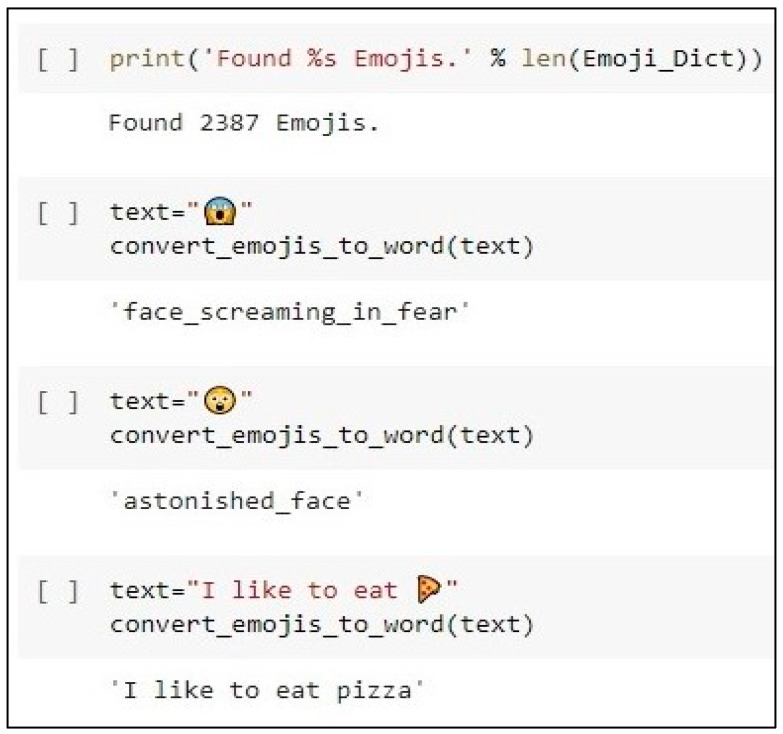
The screenshot of code of replacing emojis with words.

**Figure 11 sensors-23-01748-f011:**
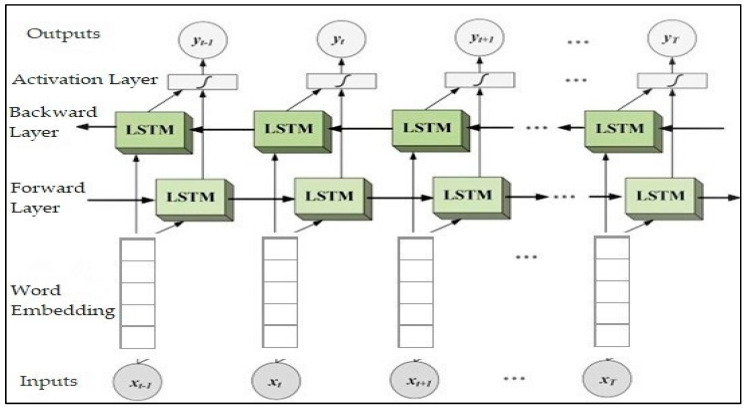
The basic architecture of Bi-LSTM uses word embedding.

**Figure 12 sensors-23-01748-f012:**
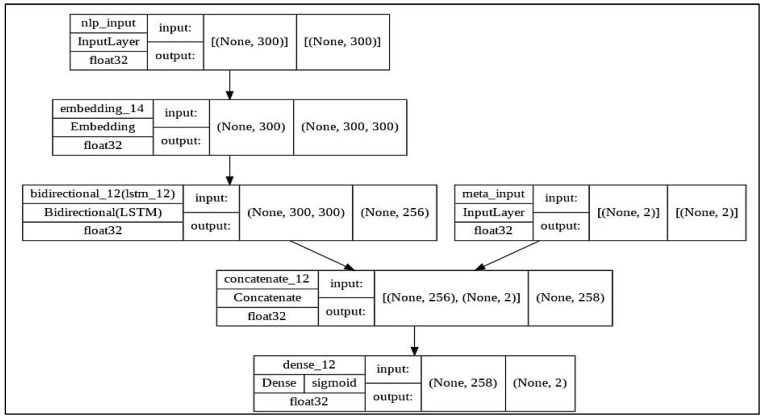
The concatenation layer in the proposed model.

**Figure 13 sensors-23-01748-f013:**
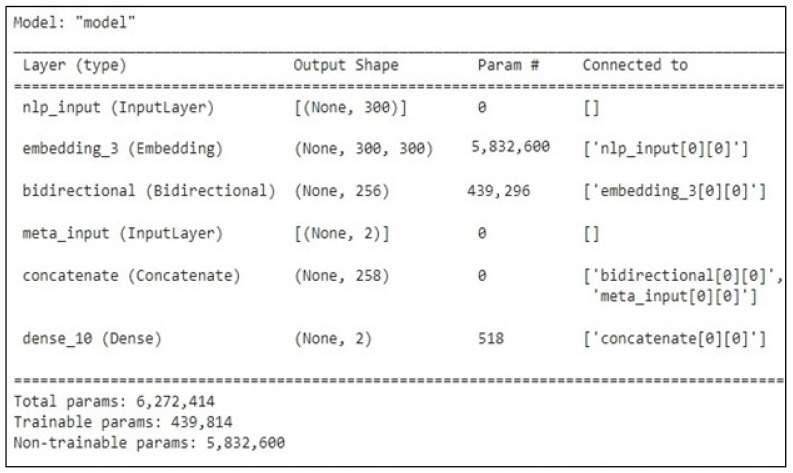
The configuration of the Bi-LSTM model.

**Figure 14 sensors-23-01748-f014:**
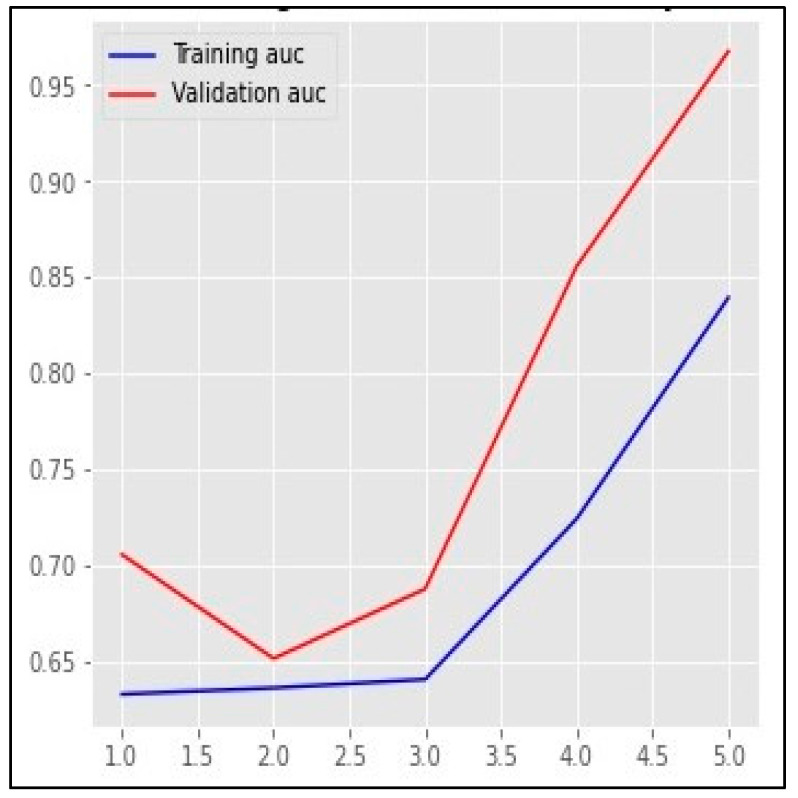
The training and validation AUC of the proposed Bi-LSTM model.

**Figure 15 sensors-23-01748-f015:**
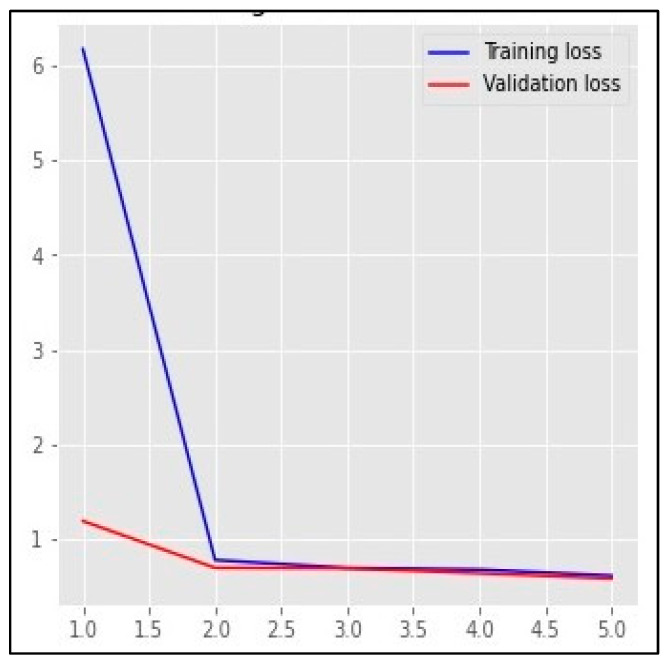
The training and validation loss of the proposed Bi-LSTM model.

**Table 1 sensors-23-01748-t001:** Use social networks and messenger services worldwide in January 2021, with details for mobile social media use, according to the Digital 2021 Global Digital Overview report [48].

Details Related to Social Media Users	Number or Percentage
Total number of active social media users	4.20 Billion
Social media users as a percentage of the global population	53.60%
Annual change in the number of global social media users	+13.2% (+490 Million)
Total number of social medias accessing via mobile phones	4.15 Billion
Percentage of total social media users accessing via mobile	98.80%

**Table 2 sensors-23-01748-t002:** The description of the Fakeddit dataset.

Feature	Description
Author	A person or bot that registers on social networking sites.
Title	The title or brief description of the post.
Domain	The source of news.
Has_image	The post whether attached with an image or not.
Id	The ID of the post.
Num_comments	The number of comments on the post; this feature represents the level of popularity of the post.
Score	The numerical score another user gives to the post; This feature shows if another user approves or declines to share the post.
Upvote_ratio	A value representing an estimate of the approval/disapproval of other users on posts.
Body	The content of the comment.

**Table 3 sensors-23-01748-t003:** The emotion groups are divided based on analysis of crowd comments’ emotions and their association with fake and real news.

Group Name	Emotion Categories of the Group	Normalized Label
Novelty	Fear, disgust, surprise	1
Expectation	Anticipation, sadness, joy, trust	0
Neutral	(Fear, disgust, surprise) = (anticipation, sadness, joy, trust)	0.5

**Table 4 sensors-23-01748-t004:** The architecture of the proposed fake news detection model.

No.	Structure or Hyperparameter Name	Type or Value
1	Layers	Embedding layer Bidirectional LSTM layer Dense layer
2	Word embedding dimension	300
3	No. of hidden states	128
4	Dropout	0.2
5	Recurrent dropout	0.2
6	Regularizer L1	0.0001
7	Regularizer L2	0.001
8	Activation function	Sigmoid
9	Loss function	Binary_Crossentropy
10	Optimizer	Adam
11	Learning rate	0.1
12	Batch size	256
13	No. of epochs	5

**Table 5 sensors-23-01748-t005:** The results of the proposed models according to the features used, (A checkmark indicates that this feature is selected).

Model	Textual Content Features (News Titles)	Features Based on Titles’ Sentiment	Features Based on Comments’ Emotions	AUC	F1-Score
LSTM	√			89.99%	90.98%
LSTM	√	√	√	90.16%	91.78%
GRU	√			91.65%	92.23%
GRU	√	√	√	92.60%	94.09%
CNN	√			94.14%	96.39%
CNN	√	√	√	96.05%	97.76%
BI-LSTM	√			94.65%	95.54%
BI-LSTM	√	√	√	96.77%	97.81%

**Table 6 sensors-23-01748-t006:** The features used in detection models, (A checkmark indicates that this feature is selected).

Study	Model	Textual Content	Visual Content	Social-Based Features	Metadata-Based Features	Comments-Based Features	Emotions-Based Features	Sentiments-Based Features
Nakamura, Levy [76]	BERT + ResNet50	√	√	√	√	√		
Kaliyar, Kumar [77]	DeepNet	√		√				
Kirchknopf, Slijepčević [78]	Multimodal architecture (BERT + CNN + MLP)	√	√	√	√	√		
Xie, Liu [79]	SERN model based on (BERT + ResNet + MLP)	√	√			√		
Raza and Ding [80]	FND-NS model based on BART	√		√	√	√		
Our Proposed Model	Bi-LSTM model	√				√	√	√

**Table 7 sensors-23-01748-t007:** The results of the detection models for the benchmarking dataset.

Study	AUC	Accuracy	F1-Score
Nakamura, Levy [76]	-	86.54%	-
Kaliyar, Kumar [77]	-	86.4%	87.2%
Kirchknopf, Slijepčević [78]	-	94.4%	-
Xie, Liu [79]	-	96.63%	96.63%
Raza and Ding [80]	70.4%	74.8%	74.9%
Our proposed model	96.77%	96.89%	97.81%

## Data Availability

Dataset: https://github.com/entitize/Fakeddit (accessed on 22 December 2022). The polarity result of emotion analysis: https://drive.google.com/drive/folders/147bXzQuMfIxE8Wvnw03wCVaB4Jp_l3qt?usp=sharing (accessed on 22 December 2022).

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
