# Peer review of "Fake News Detection Model on Social Media by Leveraging Sentiment Analysis of News Content and Emotion Analysis of Users’ Comments"

_sensors, 2023, doi:10.3390/s23041748_

Round 1
Reviewer 1 Report
Many figures need a higher quality.
I do not understand why emotions are often introduced in round brackets within the discourse.
Each macro section, e.g., 1,2,3,4,5, needs an introduction of the different subsections at the beginning, e.g., "section 2.1 talks about ..., section 2.2 on the other hand deals with the topic of..." and so on.
Introduction section 1 generally does not need subsections, but stops at the description of how the subsequent sections are made. Therefore I would move 1.2, 1.3, 1.4 and 1.5 within section 2 and reorganize the latter. You should also take into account recent work using architectures based on transformers, such as https://doi.org/10.3390/s21010133.
Also in the introductory section I would try to better highlight the contributions of this article, perhaps with a bulleted list.
Furthermore, it would be important to specify the libraries used (keras? torch?), possibly the IDE if in addition to Colab, and find a strong justification for the use of NN models based on Bi-LSTM rather than Transformers.
Reviewer 2 Report
This paper extracted features from sentiment analysis of news articles and emotion analysis of users' comments regarding the news. The authors use these features and news content feature and propose Bidirectional Long-Short Term Memory model to detect fake news. They use the standard Fakeddit dataset to train and test the proposed model. The experiments show that the proposed model can produce a high detection accuracy as 96.77% of the Area under the ROC Curve measure. The topic of the paper is interesting however the English of the paper should be improved with professional English proofreading. For instance, "The results prove that the features extracted based on sentiment analysis of news, which represents the publisher's stance, and emotion analysis of comments, which represent the crowd's stance, contribute to raising the efficiency of the detection model." There are too many "which" in this sentence. Also, the authors should cite the relevant papers published recently in this area in the following.
Qiang Wang, Wen Zhang, Jian Li, Feng Mai, Zhenzhong Ma.: Effect of online review sentiment on product sales: The moderating role of review credibility perception. Computers in Human Behavior, Volume 133, August 2022, 107272.
Author Response
Please the attachment

Reviewer 3 Report
The paper is really interesting and well-written.
There are some points that need to be addressed.
1. Introduction is too long and contains a significant part of the literature review. In my opinion, it must be split into two sections. From the point "1.2. Fake News Overview" make the second section. maybe Fake news and social media.
2. Rename the section "Related Works"
3. In the conclusion section please write Academic implications, practical implications, and the limitations of the study.
4. In the paper you use figures from other studies. I am not sure that you don't need written permission. Please check.
Author Response
Please the attachment

Round 2
Reviewer 2 Report
The authors has addressed all the concerns raised by the reviewers in the previous version. I have no further comments on the paper. However, Figures 11 and 14 are not clear and they need to be redrawn like Figure 10.
Author Response
The Figures 11 and 14 have been modified
Reviewer 3 Report
Thank you for taking into consideration my comments.
Author Response
I am very grateful to you for your valuable comments that have improved the article